# Point2SSM: Learning Morphological Variations of Anatomies from Point Clouds

**Jadie Adams & Shireen Y. Elhabian**
Scientific Computing and Imaging Institute
Kalhert School of Computing
University of Utah, USA
`{jadie,shireen}@sci.utah.edu`

## Abstract

We present Point2SSM, a novel unsupervised learning approach for constructing correspondence-based statistical shape models (SSMs) directly from raw point clouds. SSM is crucial in clinical research, enabling population-level analysis of morphological variation in bones and organs. Traditional methods of SSM construction have limitations, including the requirement of noise-free surface meshes or binary volumes, reliance on assumptions or templates, and prolonged inference times due to simultaneous optimization of the entire cohort. Point2SSM overcomes these barriers by providing a data-driven solution that infers SSMs directly from raw point clouds, reducing inference burdens and increasing applicability as point clouds are more easily acquired. While deep learning on 3D point clouds has seen success in unsupervised representation learning and shape correspondence, its application to anatomical SSM construction is largely unexplored. We conduct a benchmark of state-of-the-art point cloud deep networks on the SSM task, revealing their limited robustness to clinical challenges such as noisy, sparse, or incomplete input and limited training data. Point2SSM addresses these issues through an attention-based module, providing effective correspondence mappings from learned point features. Our results demonstrate that the proposed method significantly outperforms existing networks in terms of accurate surface sampling and correspondence, better capturing population-level statistics. The source code is provided at `https://github.com/jadie1/Point2SSM`.

## 1 Introduction

Statistical shape modeling (SSM) enables quantifying and characterizing the morphological variations in a group of shapes. SSM captures the inherent characteristics of a shape class or the underlying parameters that remain when global geometric information (i.e., location and orientation) is factored out (Kendall, 1977). It is a powerful tool in medical research, enabling population-level analysis of anatomies such as bones or organs, revealing correlations between shape variations and clinical outcomes. Despite being successfully applied in a wide range of tasks (including pathology detection (Atkins et al., 2017; Bischoff et al., 2014; Wang et al., 2015), disease biomarker identification (Bruse et al., 2016; Cates et al., 2014; Mendoza et al., 2014; Merle et al., 2014), surgical/treatment planning (Bhalodia et al., 2020; Carriere et al., 2014; Zhang et al., 2015; Zachow, 2015), and implant designs (Zadpoor & Weinans, 2015)), practical limitations have prevented its widespread adoption. The conventional SSM approach entails analyzing a group of *complete* surface representations obtained from 3D medical images (e.g., CT or MRI) in the form of binary volumes or meshes. Correspondence-based SSM establishes sets of geometrically and semantically consistent landmarks or correspondence points on the shape surfaces. Various optimization schemes have automated the correspondence point generation process (Davies et al., 2002; Ovsjanikov et al., 2012; Styner et al., 2006) to provide dense sets of correspondence points for each shape, including particle-based shape modeling (PSM) (Cates et al., 2007; 2017). However, such optimization techniques have *three significant limitations*. Firstly, they require *complete* shape representations in the form of high-resolution meshes or binary volumes that are *free from noise and artifacts*, which are difficult to acquire, prohibiting many use cases. Secondly, the optimization process is

time-consuming and must be performed on the entire cohort simultaneously, which greatly *hinders inference* when adding a new shape to the SSM. Lastly, these methods utilize metrics such as Gaussian entropy (Cates et al., 2007) or parametric representations (Ovsjanikov et al., 2012; Styner et al., 2006) to define optimization objectives, which may *bias or restrict* the types of variation captured by the SSM (i.e., via enforcing linearity or inheriting the topology of the pre-defined template). Deep learning methods have been proposed to predict SSM from meshes (Bastian et al., 2023; Lüdke et al., 2022; Iyer & Elhabian, 2023), addressing some of these limitations. However, such approaches rely on mesh connectivity, enforcing the same input restrictions as optimization-based methods.

Point cloud deep learning has recently shown success in tasks such as unsupervised representation learning, shape generation, point cloud up-sampling and completion, and point-to-point matching (Fei et al., 2022; Xiao et al., 2023; Akagic et al., 2022). Point clouds can be readily obtained from full-shape segmentations such as meshes when available. They can, however, also be obtained from more lightweight shape acquisition methods, e.g., thresholding clinical images, anatomical surface scanning, and combining 2D contour representations (Timmins et al., 2021; Treleaven & Wells, 2007). Thus, generating SSM directly from point clouds would significantly expand the potential clinical use cases of shape analysis. Recently, Adams & Elhabian (2023b) demonstrated that existing point cloud encoder-decoder-based completion networks perform reasonably well at SSM generation out-of-the-box. Although such architectures were not designed for correspondence tasks, the bottleneck captures a population-specific shape prior, and the continuous-mapping decoder results in ordered output, providing correspondence as a by-product. However, such methods have not been benchmarked against point correspondence approaches or tested for robustness to noise, partiality, or sparse input (Adams & Elhabian, 2023b). A myriad of challenges accompanies applying point cloud deep learning approaches to anatomical SSM - most notably, the issue of data scarcity. Deep network training requires a large cohort of representative anatomies defined from volumetric medical images, which is difficult to acquire, especially if modeling an uncommon disease or pathology. In addition, point cloud shape representations obtained from medical images may suffer from various issues, such as noise from the acquisition process, missing regions outside the scanner field of view, or sparse point clouds due to low image resolution (Razzak et al., 2018).

In this paper, we introduce Point2SSM, an unsupervised deep learning framework for learning correspondence-based SSM of anatomy directly from point clouds. Point2SSM overcomes the limitations of existing optimization-based SSM methods and point cloud networks by providing a data-driven solution that operates on unordered point clouds and infers SSMs that capture population-level statistics with good surface sampling. Point2SSM removes biases imposed by optimization assumptions and significantly relaxes the input shape requirement. Moreover, a trained Point2SSM provides a fast and efficient way to predict SSM from a new unseen point cloud without re-optimization. We benchmark existing state-of-the-art (SOTA) point cloud networks against the proposed method for the SSM application. This benchmark reveals that although these methods achieve some success on this new task, they have limited ability to address the challenges that arise in clinical scenarios. We show that Point2SSM is more robust to a limited training budget as well as to sparse, noisy, and incomplete input than existing point methods and achieves similar statistical compactness to an optimization-based SSM method. Our contributions can be summarized as follows:

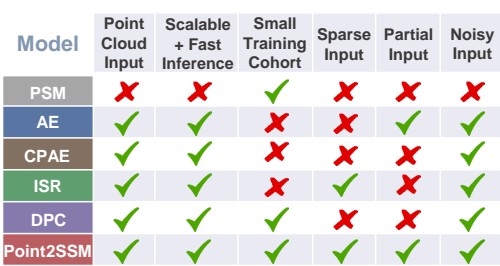

| Model | Point Cloud Input | Scalable + Fast Inference | Small Training Cohort | Sparse Input | Partial Input | Noisy Input |
|---|---|---|---|---|---|---|
| PSM | ✗ | ✗ | ✓ | ✗ | ✗ | ✗ |
| AE | ✓ | ✓ | ✗ | ✗ | ✓ | ✓ |
| CPAE | ✓ | ✓ | ✗ | ✗ | ✗ | ✓ |
| ISR | ✓ | ✓ | ✗ | ✓ | ✗ | ✓ |
| DPC | ✓ | ✓ | ✓ | ✗ | ✗ | ✓ |
| Point2SSM | ✓ | ✓ | ✓ | ✓ | ✓ | ✓ |

Figure 1: Comparison of particle-based modeling (PSM) (Cates et al., 2017), point autoencoders (AE) (Achlioptas et al., 2018), canonical point autoencoder (CPAE) (Cheng et al., 2021), Chen et al. (2020) (ISR), deep point correspondence(DPC) (Lang et al., 2021), and Point2SSM.

(1) We introduce Point2SSM, a novel point cloud approach for anatomical SSM that removes the limitations of optimization-based SSM generation techniques without hindering statistical accuracy.

(2) We provide the first benchmark of SOTA point cloud networks on the anatomical SSM task.

(3) We demonstrate Point2SSM outperforms existing methods in terms of surface sampling and correspondence accuracy and is more robust to limited data and sparse, noisy, or incomplete input.

## 2 RELATED WORK

### 2.1 OPTIMIZATION-BASED SSM

Correspondence-based SSM optimization techniques constrain points to the shape surfaces and establish intersubject correspondence via metrics such as entropy (Cates et al., 2007; 2017; Oguz et al., 2016) or minimum description length (Davies et al., 2002), or via parametric representations (Ovsjanikov et al., 2012; Styner et al., 2006). These approaches require *complete, faultless* shape inputs (surface meshes or binary segmentations) and operate on the entire cohort simultaneously, preventing the addition of a new shape without rerunning the optimization process. Convolutional deep learning approaches for predicting SSMs directly from unsegmented images have been proposed to reduce these burdens (Bhalodia et al., 2018; Adams et al., 2020; Adams & Elhabian, 2022; 2023a; Bhalodia et al., 2021; Ukey & Elhabian, 2023; Tóthová et al., 2020). However, such approaches are supervised and thus require a traditional optimization scheme for generating a training data cohort, limiting their accuracy potential based on the training set.

### 2.2 DEEP LEARNING ON POINT CLOUDS

PointNet (Qi et al., 2017a) was the first deep network designed to process raw point clouds. It employs multi-layer perceptrons (MLPs) and symmetric aggregation functions to learn permutation invariant features. PointNet++ (Qi et al., 2017b) extended this architecture to have a hierarchical structure for learning multiscale geometric information. Dynamic graph convolution (DGCNN) (Wang et al., 2019) utilized nearest neighbors to construct point cloud graphs and apply edge convolution. Transformer-based methods (i.e., PointTransformer (Zhao et al., 2021) and Point-Bert (Yu et al., 2022)) have also been developed, which apply self-attention to 3D point cloud processing to learn underlying structure. To date, there is no ubiquitous 3D backbone for point cloud networks, but the aforementioned networks are common (Xiao et al., 2023). Supervised training of point cloud networks requires large-scale, densely labeled datasets. This annotation burden has inspired the exploration of unsupervised learning of robust feature representations (Xiao et al., 2023). Unsupervised tasks include self-reconstruction, point cloud up-sampling, and completion of partial point clouds. Achlioptas et al. (2018) proposed the first point cloud autoencoder (AE) and demonstrated the generative power of the learned latent space, which led to the subsequent development of generative architectures (Xiao et al., 2023). Point completion networks have widely adopted an encoder and coarse-to-fine decoder architecture, allowing the network to learn general shape first and then refine it (Yuan et al., 2018; Fei et al., 2022). Point cloud encoder-decoder-based networks such as these have been shown to perform reasonably well at SSM generation (Adams & Elhabian, 2023b). Although these methods were not designed for SSM, the continuous mapping from the learned latent space to output space results in decoded ordered point clouds, providing correspondence as a by-product. However, such methods require a sufficiently large and representative training dataset and have yet to be compared to point correspondence approaches (Adams & Elhabian, 2023b).

### 2.3 LEARNING 3D POINT CLOUD DENSE CORRESPONDENCE

While point cloud learning for anatomical SSM is largely unexplored, point networks have been developed to establish shape correspondence. This task has been approached in a supervised manner through point cloud registration (Choy et al., 2019; Chen et al., 2019a; Gojcic et al., 2019; Huang et al., 2017), via unified embeddings of multiple shape representations (Muralikrishnan et al., 2019), and via part labels (Bhatnagar et al., 2020). Recently, unsupervised methods have been developed that formulate shape correspondence from either a *pairwise* or class-level, *global* standpoint. Pairwise approaches seek to find a point-to-point mapping from a source shape to a target shape. Mesh-based methods have been established for this task using functional maps (Ovsjanikov et al., 2012; Donati et al., 2020; Ginzburg & Raviv, 2020; Eisenberger et al., 2020), which require connectivity. Functional maps have been extended to point clouds, using spectral matching to define correspondence (Marin et al., 2020). Other approaches extract correspondence by learned deformations to a predefined template (Deprelle et al., 2019; Cosmo et al., 2016). Recent approaches utilize deep networks to learn a matching permutation between a source and target point cloud. Zeng et al. (2021) utilize an encoder-decoder architecture to regress the shape coordinates for permuted reconstruction, whereas Lang et al. (2021) opt to drop the decoder and use the original point cloud in reconstruction, achieving better performance by leveraging similarity in the learned feature space.

Prior global correspondence approaches have focused on discovering class-specific keypoints (a.k.a. ordered stable interest or structure points) from point clouds (Suwajanakorn et al., 2018; Fernandez-Labrador et al., 2020; Jakab et al., 2021). These semantically consistent points are typically sparse. More relevant works have sought to define **dense** keypoints or structure points - a task highly related to correspondence-based SSM. Chen et al. (2020) developed a network that reconstructs point clouds in a consistent way using an input subset and learned features, providing a correspondence model with meaningful principal component analysis embedding. Liu & Liu (2020) leveraged part features learned by a branched AE (Chen et al., 2019b) to establish intraclass correspondence. However, this approach requires additional knowledge of the shape surface to compute the occupancy for training the implicit function. Cheng et al. (2021) developed a self-supervised canonical point autoencoder that utilizes mapping to a canonical primitive (sphere) to establish order across classes of shapes. While these methods establish correspondence, they haven't been tested against anatomical SSM challenges, such as limited training budget or input noise, missingness, or sparsity.

## 3 METHODS

### 3.1 PROPOSED APPROACH: POINT2SSM

Let $\mathbb{S}$ denote an unordered point cloud of $P$ points representing an anatomical shape: $\mathbb{S} = \{\mathbf{s}_1, \ldots, \mathbf{s}_P\}$ with $\mathbf{s}_i \in \mathbb{R}^3$. Given a subset of $N$ points in $\mathbb{S}$, the goal of Point2SSM is to predict a set of $M$ correspondence points, denoted $\mathbb{C}$. Point2SSM learns correspondence in a self-supervised manner by estimating a set of points $\mathbb{C}$ that best reconstructs the full point clouds $\mathbb{S}$. It is comprised of an encoder and a transformer-like attention module, as shown in figure 2.

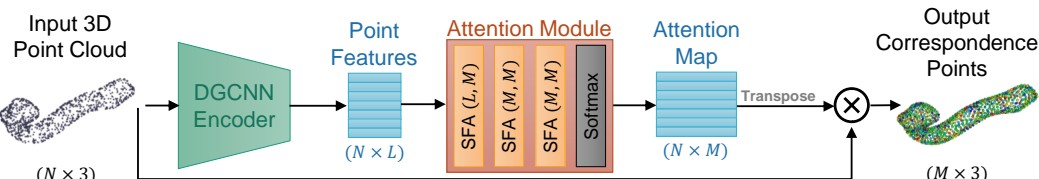

Figure 2: Point2SSM architecture

The encoder learns an $L$-dimensional feature vector for each point. We select a DGCNN (Wang et al., 2019) encoder architecture to learn topological information via edge convolution. Unlike PointNet and other methods that treat each point independently, DGCNN incorporates local neighborhood information, enriching the representational power and capturing global semantic characteristics. This formulation is well suited for learning features for anatomical SSM because anatomies share typical characteristics that stem from the underlying mechanisms involved in their formation.

The attention module predicts an attention map from the feature representation via self-attention. The output correspondence points are then computed via matrix multiplication between the attention map and the input point cloud. Similar to existing methods (Chen et al., 2020; Lang et al., 2021), each resulting output point is a weighted combination of all of the input points. This constraint, combined with the attention module, increases surface sampling accuracy. The attention module must capture the structural characteristic of the shape from the point features to learn correspondence. Hence, we leverage the self-feature augment (SFA) blocks introduced in Wang et al. (2022). The SFA blocks integrate the information from different point features and establish the spatial relationship among points by introducing self-attention. In this manner, SFA captures global information and reveals detailed shape geometry, generating semantically consistent attention maps for all shapes in a cohort. Resulting attention maps are visualized in appendix B.

The Point2SSM loss function is comprised of two terms - one that encourages $\mathbb{C}$ to reconstruct $\mathbb{S}$ and one that regularizes $\mathbb{C}$, encouraging better correspondence. Chamfer distance (CD) is used to measure the difference between the point clouds in a permutation-invariant way:

$$\mathrm{CD}(\mathbb{C}, \mathbb{S}) = \frac{1}{|\mathbb{C}|} \sum_{\mathbf{c} \in \mathbb{C}} \min_{\mathbf{s} \in \mathbb{S}} ||\mathbf{c} - \mathbf{s}||_2^2 + \frac{1}{|\mathbb{S}|} \sum_{\mathbf{s} \in \mathbb{S}} \min_{\mathbf{c} \in \mathbb{C}} ||\mathbf{s} - \mathbf{c}||_2^2 \tag{1}$$

A pairwise mapping error (ME), originally proposed in (Lang et al., 2021), is adapted to provide an output regularization loss term. The ME between two output point clouds $\mathbb{C}'$ and $\mathbb{C}''$ is defined as:

$$\text{ME}(\mathbb{C}', \mathbb{C}'') = \frac{1}{M * K} \sum_{i=1}^{M} \sum_{j \in \mathcal{N}_{\mathbb{C}'}(\mathbf{c}_i')} v_{ij}' \left\| \mathbf{c}_i'' - \mathbf{c}_j'' \right\|_2^2 \tag{2}$$

where $\mathcal{N}_{\mathbb{C}'}(\mathbf{c}_i')$ are the $K$ indices of the $K$-nearest neighbors of point $\mathbf{c}_i'$ in $\mathbb{C}'$ (in Euclidean distance). Here $v_{il}' = e^{-\left\| \mathbf{c}_i' - \mathbf{c}_l' \right\|_2^2}$ weights the loss elements according to the proximity of the neighbor points. ME loss encourages the point neighborhoods in $\mathbb{C}'$ to be similar to $\mathbb{C}''$, promoting correspondence. ME is computed between all pairs of output in a minibatch of size $B$. Point2SSM is not sensitive to the choice of $B$, as demonstrated in appendix F. The Point2SSM loss is thus defined as:

$$\mathcal{L} = \frac{1}{B} \sum_{i=1}^{B} \text{CD}(\mathbb{C}^i, \mathbb{S}^i) + \alpha \left( \frac{1}{(B-1)^2} \sum_{i=1}^{B} \sum_{j=1, j \neq i}^{B} \text{ME}(\mathbb{C}^i, \mathbb{C}^j) + \text{ME}(\mathbb{C}^j, \mathbb{C}^i) \right) \tag{3}$$

where $\alpha$ is a hyperparameter that controls the effect of the regularization. An ablation experiment that demonstrates the impact of the encoder architecture, attention module architecture, and ME loss is provided in appendix A.

## 3.2 COMPARISON MODELS

We benchmark the following SOTA methods on the anatomical SSM tasks and compare their performance to that of Point2SSM:

**PSM (Cates et al., 2017)** denotes particle-based shape modeling, the SOTA optimization-based technique for generating correspondence points from complete shape surface representations (Goparaju et al., 2022). We use the mesh-based implementation of PSM available in the open-source toolkit ShapeWorks (Cates et al., 2017). We include this method to provide context regarding the expected modes of variation compactness of SSM for a given anatomy.

**PN-AE (Achlioptas et al., 2018)** is the autoencoder formulated by Achlioptas et al. (2018) with PointNet (Qi et al., 2017a) encoder. The combination of bottleneck and MLP decoder results in consistent output ordering of $\mathbb{C}$ (Adams & Elhabian, 2023b).

**DG-AE (Wang et al., 2019)** is a variant of PN-AE where the PointNet (Qi et al., 2017a) encoder is replaced with a DGCNN (Wang et al., 2019) encoder. It is included to assist in comparing the AE framework with Point2SSM and DPC.

**CPAE (Cheng et al., 2021)** is the canonical point autoencoder that maps points to a sphere template in the bottleneck and then reconstructs the points in an ordered fashion.

**ISR (Chen et al., 2020)** denotes the method for learning intrinsic structural representation (ISR) points proposed by Chen et al. (2020). ISR utilizes Chamfer loss (equation 1), a PointNet++ (Qi et al., 2017b) encoder, and an MLP point integration module that maps the features to a probability map. The probability map is multiplied by a learned subset of the input points to provide output.

**DPC (Lang et al., 2021)** is the deep point correspondence model proposed by Lang et al. (2021). DPC is a pairwise correspondence method that takes the source and target point clouds as input and outputs the source reordered to match the target. The architecture comprises a DGCNN (Wang et al., 2019) encoder and cross- and self-construction modules that utilize latent similarity. To adapt DPC to provide global correspondence, the same target point cloud or reference is used for every shape in inference. In this work, the reference is selected as the point cloud with the minimum Chamfer distance to all other point clouds in the training set.

## 3.3 EVALUATION METRICS

An ideal SSM accurately samples the shape surface via uniformly distributed points that are constrained to lie on the surface. Simultaneously, it captures anatomically relevant mappings between shapes by establishing consistent, invariant points (i.e., correspondences) across diverse populations with varying forms. Consequently, the evaluation of an SSM should encompass both aspects: the accuracy of surface sampling and the efficacy of the extracted shape statistics.

Three metrics are used to define the *point surface sampling accuracy*: $CD(\mathbb{C}, \mathbb{S})$, Earth movers distance (EMD), and point-to-face distance (P2F). EMD requires both point clouds to have the same number of points; thus, a subset of points in $\mathbb{S}$ is selected using farthest point sampling (Qi et al., 2017b). In this way, EMD captures whether the predicted correspondence points cover the entire shape. P2F distance is calculated as the distance of each point in $\mathbb{C}$ to the closest face of the ground truth mesh, indicating how well points are constrained to the surface.

In an ideal SSM, point locations and neighborhoods are preserved across all shapes in the cohort, leading to a meaningful model with compact modes of shape variation. In SSM analysis, correspondence points are averaged to generate a representative mean shape, and principal component analysis (PCA) is performed to compute the significant modes of variation. These modes can be used in medical hypothesis testing and visualized by deforming the mean shape along each basis of the linear subspace (Cates et al., 2014). Three statistical metrics are used to evaluate *SSM correspondence accuracy*: compactness, generalization, and specificity (Munsell et al., 2008). A compact SSM represents the training data distribution using the minimum number of parameters; thus, compactness is quantified as the number of PCA modes required to capture 95% of the variation in the correspondence points. Moreover, a good SSM should generalize well from training examples to unseen examples. The generalization metric quantifies how well the SSM generalizes from training examples to unseen examples via the reconstruction error ($L2$) between held-out correspondence points and those reconstructed via the training SSM. The specificity metric measures the degree to which the SSM generates valid instances of the shape class presented in the training set. It is computed as the average distance between correspondences sampled from the training SSM and the closest existing training correspondences. The equations of these metrics are available in Munsell et al. (2008) and Appendix C. Additionally, ME (Equation 2) is included as an SSM metric since it captures how well point neighborhoods correspond across the cohort.

Both categories of metrics must be used to evaluate the accuracy of SSM, as one does not imply the other. For example, a network that yields identical output given any input will perform very well on statistical metrics but poorly on sampling metrics. Conversely, a network that perfectly reconstructs the input in an unordered fashion will succeed at point sampling metrics but fail at statistical metrics.

## 4 EXPERIMENTS AND ANALYSIS

We utilize three challenging organ mesh datasets of various sample sizes to benchmark the performance of Point2SSM and the comparison methods: spleen (Simpson et al., 2019) (40 shapes), pancreas (Simpson et al., 2019) (272 shapes), and left atrium of the heart (1096 shapes). We elect to acquire point clouds from meshes to enable benchmarking against PSM, which requires mesh connectivity. The spleen dataset represents a typical SSM scenario with limited data and considerable variation in shape and curvature. The pancreas dataset consists of cancer patients, resulting in increased shape variability due to varying tumor sizes and morphologies. The left atrium is a notoriously difficult shape to model because it exhibits significant variations in volume, appendage size, and pulmonary vein configuration, number, and length. Appendix G provides a visualization of the organ cohorts. Note that the nonanatomical shape datasets previously used to benchmark comparison methods are much larger (i.e., tens of thousands).

Meshes are pre-aligned to factor out global geometric information via iterative closest points (Besl & McKay, 1992) utilizing the ShapeWorks (Cates et al., 2017) toolkit. The aligned, unordered mesh vertices serve as ground truth complete point clouds, $\mathbb{S}$. In all experiments, we set $N = 1024$, $L = 128$, $M = 1024$, and batch size $B = 8$, unless otherwise specified. Point2SSM is not sensitive to batch size (appendix F). During training, input point clouds are generated by randomly selecting $N$ points from $\mathbb{S}$ each iteration and uniformly scaling them to be between -1 and 1. The datasets are randomly split into a training, validation, and test set using an 80%, 10%, 10% split. Adam optimization with a constant learning rate of 0.0001 is used, and model training is run until convergence via validation assessment. Specifically, a model is considered converged if the validation CD has not improved in 100 epochs. Models resulting from the epoch with the best validation CD are used in the evaluation. A 4x TITAN V GPU was used to train all models. For Point2SSM loss (equation 3), $\alpha$ is set to 0.1 based on tuning using the validation set. For comparison models, the originally proposed loss is used with the reported tuned hyperparameter values. Appendix D provides all parameters and appendix E provides a comparison of model memory footprint.

## 4.1 RESULTS

Figure 3 provides an overview of the results on all datasets. Point2SSM significantly outperforms the existing point methods on the surface sampling metrics. This is further illustrated in figure 4, which provides a point-level visualization of the test example with median P2F distance output by each model. Figure 3 additionally demonstrates that Point2SSM performs comparably with respect to statistical metrics to PSM and provides the best compactness on the left atrium dataset. Of the point-based methods, Point2SSM achieves the best compactness on all datasets - with the exception of the spleen DG-AE , which greatly suffers in terms of point sampling metrics. The AE methods aggregate features into a global ($L \times 1$) feature in the bottleneck. This restriction enforces a shape prior, providing compactness, but it greatly limits model expressivity, hindering accurate surface sampling. The CPAE output reconstructs the point cloud to a degree but does not provide correspondence. The resulting CPAE SSM is not compact or interpretable, likely because learning a canonical mapping is too complex given a small training set. Like Point2SSM , ISR and DPC predict output as a weighted combination of input points, thus providing good performance on surface sampling metrics. However, they do not provide SSM that is as compact, generalizable, or specific as Point2SSM. While ISR learns to map a learned subset of input points to correspondence points via a point integration model, Point2SSM makes use of the full input information. Additionally, it suffers from its individual treatment of points, whereas Point2SSM incorporates neighborhood information via edge convolution. DPC is comprised of only an encoder and utilizes latent similarity alone to compute an affinity matrix for establishing correspondence. In contrast, Point2SSM learns correspondence in a global manner by incorporating an attention module, which allows for capturing population shape statistics without bias induced by reliance on a reference shape. Point2SSM combines the strengths of the comparison methods, providing the best overall accuracy.

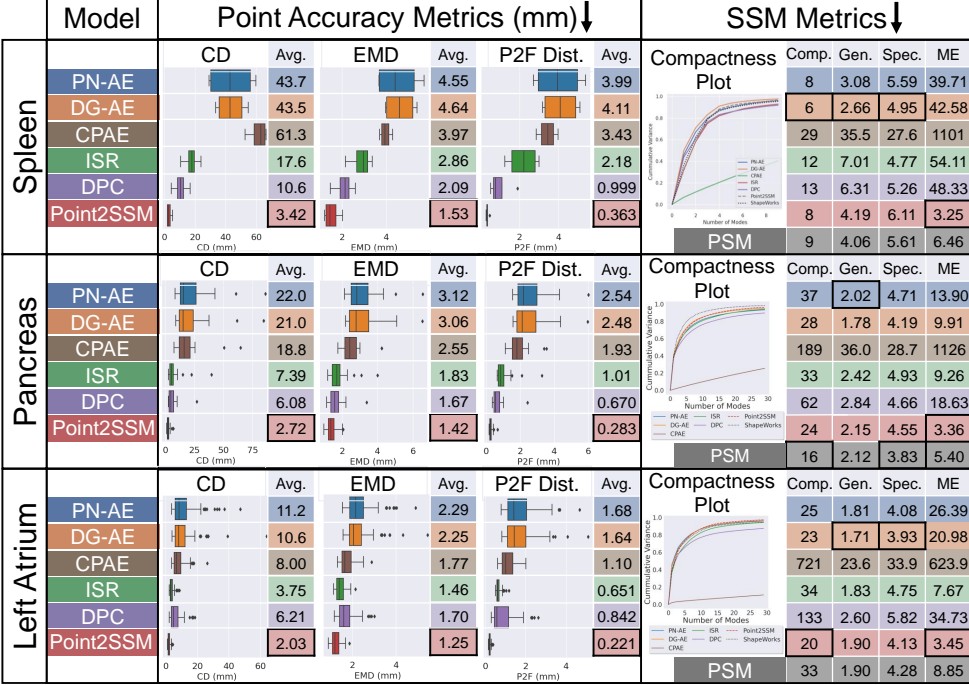

Figure 3: Accuracy metrics are reported with best values outlined. Boxplots show the distribution across test sets and averages are reported to the right. Compactness plots show cumulative population variation captured by PCA modes, larger area under the curve indicates a more compact model.

Figure 5 displays the SSM resulting from Point2SSM on the pancreas dataset. The mean shape is plausible and interpretable, and the individual point locations are geometrically consistent across shapes. The primary modes of variation in the pancreas are semantically similar to those resulting from the PSM method, suggesting they accurately capture population statistics and could be similarly used in downstream tasks. Evaluation of one such task (pancreatic tumor classification) is provided in appendix H, demonstrating the superior predictive power of Point2SSM output.

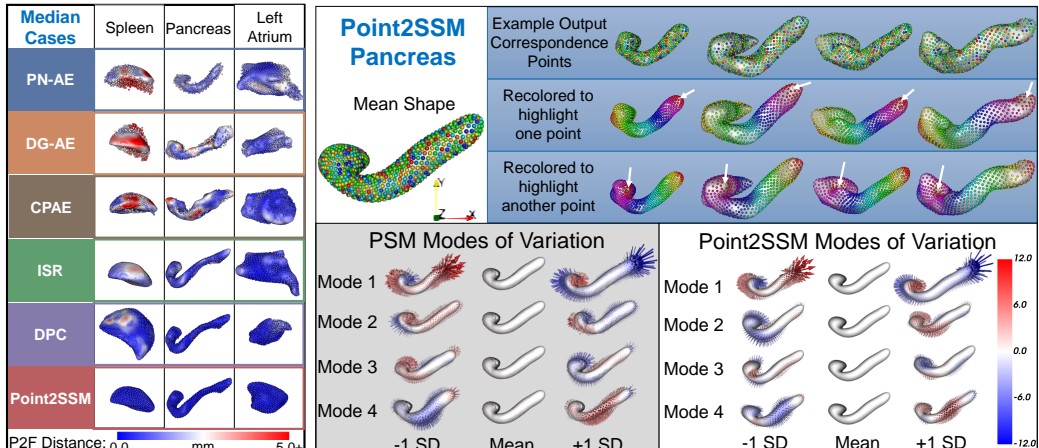

Figure 4: The test examples with median P2F distance output from each model are shown over ground truth meshes. P2F distance is displayed via a color map.

Figure 5: The pancreas SSM from Point2SSM is displayed. Point color denotes correspondence. Recoloring according to the distance to a selected point is provided for further illustration. The first four modes of variation are shown for the PSM and Point2SSM model at ±1 standard deviation from the mean. The heatmap and vector arrows display the distance to the mean.

Appendix I displays the first two modes of variation captured by Point2SSM and comparison models on the spleen and left atrium. Point2SSM is the only point-based method that provides similar, if not more smooth and interpretable, modes to the PSM method. Not only does Point2SSM allow for faster inference than PSM, but it is also more scalable. While PSM accuracy is not largely affected by the shape cohort size, the optimization process is much slower given a large cohort. Fitting the ShapeWorks PSM model to the large left atrium cohort required running optimization incrementally over four days, whereas, the Point2SSM model required under eight hours to train on a GPU.

## 4.2 ROBUSTNESS EVALUATION

We perform experiments to demonstrate model robustness using the medium-sized pancreas dataset. These results are plotted in figure 6. In all robustness experiments, the training and testing point clouds are corrupted in the same manner. Appendix J provides a visualization of example input.

**Robustness against input noise.** To analyze the effect of noise on performance, we apply random Gaussian noise to input point clouds at various levels: 0.25, 0.5, 1, and 2 mm standard deviation. Point2SSM and the comparison models are not significantly impacted by input noise, and Point2SSM achieves the best accuracy at all noise levels. These results indicate that by adding the denoising task in training, Point2SSM can easily be made robust to input noise.

**Robustness against partial input.** To analyze the impact of input with missing regions, we remove continuous regions at random locations of various sizes (5%, 10%, and 20% of the total points) from the input. The CD and EMD metrics capture how well the output fills in the missing regions to provide full coverage. The autoencoder architectures (Æ and DG-AE) are the least impacted by partial input, which is logical given this is the architecture used in point completion networks. Not only does Point2SSM perform similarly or better than all models regarding the distance metrics, but it also preserves compactness better than DPC and ISR with increasingly partial input.

**Robustness against sparse input.** To test the impact of input density, we train the models with input size $N$ of 128, 256, 512, 1024, 2048, and 4096, keeping $L$ fixed at 128 and output size $M$ fixed at 1024. This experiment benchmarks the network's ability to both upsample and provide SSM. The CPAE model and DPC model require $N = M$ and are thus excluded. Point2SSM achieves the best overall accuracy; however, performance declines given very sparse input ($N = 128$).

**Impact of training sample size.** The final experiment benchmarks the effect of training size on model performance. We consistently define random subsets of the 216 training point clouds of size

100, 50, 25, 12, and 6. DPC and Point2SSM perform the best on the distance metrics, demonstrating impressive robustness (generalizing to the test set with only six training examples). Compactness results are excluded since compactness depends on the variation in the training cohort.

The robustness experiments demonstrate that Point2SSM combines the strengths of all existing models. It performs as well as Æand DG-AE given large missing regions and as well as DPC given a small training cohort. These comparisons are compiled in figure 6.

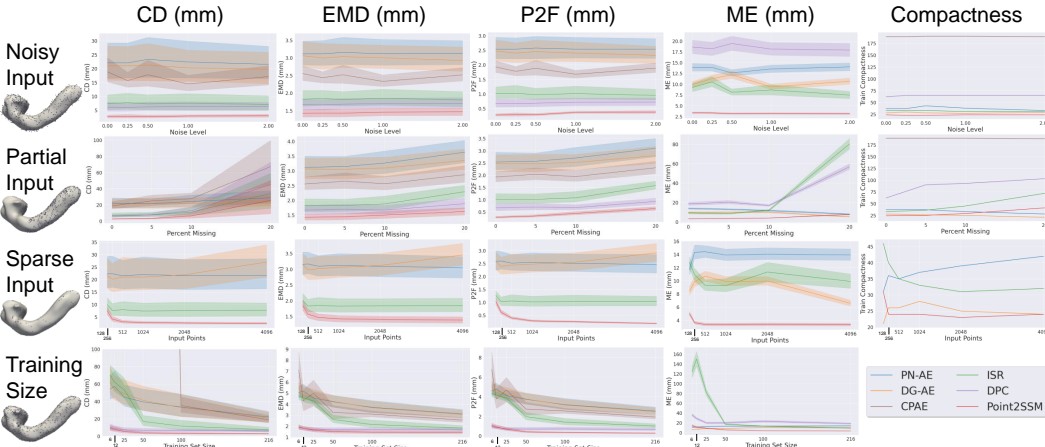

Figure 6: Robustness experiment results on the pancreas test set. Distance metrics are shown with standard deviation error bands. Compactness is calculated at 95% variability.

### 4.3 LIMITATIONS AND FUTURE DIRECTIONS

Deterministic deep learning frameworks like Point2SSM have some limitations. They produce overconfident estimates that could potentially misrepresent shapes, especially when dealing with noisy, partial, and sparse inputs. Incorporating uncertainty quantification would enhance the reliability of deploying Point2SSM in sensitive clinical decision-making scenarios. Additionally, Point2SSM requires roughly aligned input point clouds and is designed to produce SSM of a single anatomy (i.e., bone or organ). While Point2SSM can be trained on multiple anatomies (see appendix L), multi-anatomy training does not notably improve accuracy. Broadening the scope of Point2SSM to handle misalignment and to leverage multiple anatomies directly in training would increase applicability.

### 5 CONCLUSION

We introduced Point2SSM, the first deep learning method designed to produce 3D anatomical SSM directly from point clouds in an unsupervised manner. Point2SSM provides substantial advantages over traditional SSM generation approaches, offering a data-driven solution free of biases stemming from reference selection, metrics, or parametric representations used in classical methods. By reducing the input requirement from complete, noise-free shape representations to point clouds, Point2SSM significantly broadens the potential use cases of SSM. Additionally, the scalability and fast inference distinguish Point2SSM from optimization-based SSM generation methods, which are slow given large cohorts and necessitate complete reoptimization to incorporate new shapes. Deep learning approaches such as Point2SSM also enable incremental model updating through sequential or online learning. This adaptability is crucial to real-world clinical scenarios where shape data accumulates over time. Furthermore, our approach enables concurrent SSM prediction and supplementary tasks such as up-sampling, completion, or noise removal. Traditional methods cannot be applied when dealing with sparse, partial, noisy observations. In rigorous evaluations, Point2SSM outperforms state-of-the-art point cloud networks in surface sampling and correspondence accuracy. Moreover, it exhibits robustness in challenging clinical modeling situations, deftly managing limited data and handling noisy, incomplete, and sparse shape representations. Ultimately, our proposed methodology enhances the feasibility of SSM generation and broadens its possible applications, potentially accelerating its adoption as an invaluable tool in clinical research.

ACKNOWLEDGMENTS

This work was supported by the National Institutes of Health under grant numbers NIBIB-U24EB029011 and NIAMS-R01AR076120. The content is solely the responsibility of the authors and does not necessarily represent the official views of the National Institutes of Health. The authors would like to thank the University of Utah Division of Cardiovascular Medicine for providing left atrium MRI scans and segmentations from the Atrial Fibrillation projects.

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

## A  POINT2SSM ABLATION EXPERIMENT

We perform an ablation experiment on the pancreas dataset to analyze the impact of each aspect of the Point2SSM model. To illustrate the impact of the DGCNN (Wang et al., 2019) encoder, we design a PointSSM variant with the PointNet (Qi et al., 2017a) decoder used in PN-AE . The Point2SSM attention module (denoted ATTN) is comprised of attention-based SFAWang et al. (2022) blocks. To analyze this impact, we design a variation of Point2SSM where the attention module is replaced with an MLP-based architecture. For the MLP-based architecture, we elect to use the Point Integration Module proposed in ISR , with three 1D convolution blocks. Finally, to test the impact of the ME loss in equation 3, we test without it by setting $\alpha = 0$.

The results of this ablation are shown in table 1, with the full proposed Point2SSM in the final row. The DGCNN-encoder and ATTN attention module both provide surface sampling and correspondence accuracy improvements. The addition of the ME loss ($\alpha = 0.1$) improves the correspondence accuracy without reducing the surface sampling accuracy.

Table 1: Point2SSM ablation experiment on the pancreas dataset. Average test set values are reported for point accuracy distance metrics in mm. SSM metrics are calculated at 95% variability.

| Pancreas Point2SSM Ablation | | | Point Accuracy Metrics (mm) ↓ | | | SSM Metrics ↓ | | |
|---|---|---|---|---|---|---|---|---|
| Encoder | Attention Module | $\alpha$ | CD | EMD | P2F | Comp. | Gen. | Spec. |
| PointNet | MLP | 0 | 7.35 | 1.78 | 0.833 | 52 | 2.96 | 4.48 |
| PointNet | ATTN | 0 | 3.00 | 1.44 | 0.306 | 27 | 2.24 | 4.67 |
| DGCNN | MLP | 0 | 3.40 | 1.46 | 0.378 | 31 | 2.2 | 4.52 |
| DGCNN | ATTN | 0 | 2.87 | 1.43 | 0.283 | 26 | 2.32 | 4.80 |
| DGCNN | ATTN | 0.1 | 2.72 | 1.42 | 0.283 | 24 | 2.15 | 4.55 |

# B ATTENTION MAP VISUALIZATION

Figure 7 illustrates the attention map weights learned by the Point2SSM attention module on the pancreas dataset. Output correspondence points are a weighted combination of the input points, where the learned attention map defines the weights. Figure 7 highlights two output correspondence points across shapes. The attention maps show the weights on the input points (via color map) that generated the selected output point. The maps illustrate which input points were most important for defining a given output point. Note the maps highlight similar anatomical regions across samples for a given output corresponding point.

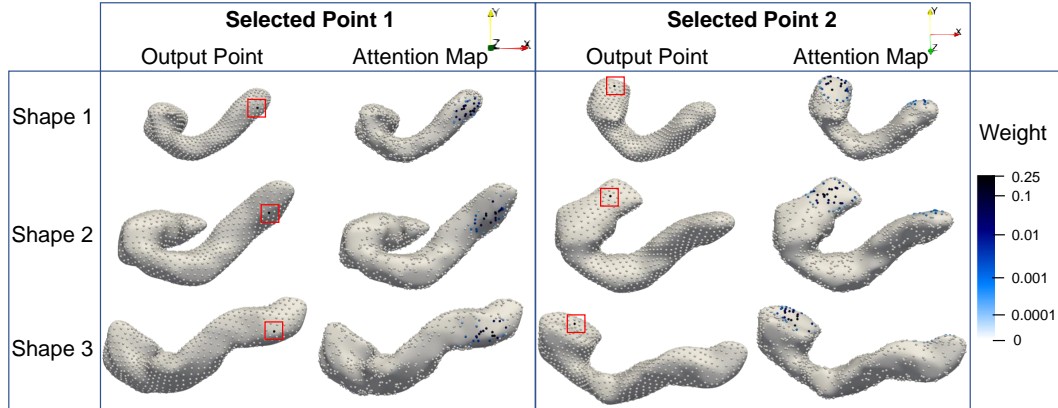

Figure 7: Two output points (highlighted in red boxes) across three pancreas shapes are shown with the corresponding weights (blue log scale color map) on the input point clouds.

# C SSM EVALALUATION METRICS

As is standard (Munsell et al., 2008), we utilize three statistical metrics to evaluate correspondence accuracy: compactness, generalization, and specificity. A compact SSM represents the training data distribution using the minimum number of parameters. We quantify **compactness** as the number of PCA modes required to capture 95% of the total variation in the output training cohort correspondence points, where fewer modes indicate a more compact model. The compactness plots in figure 3 show the cumulative explained variance as the number of modes increases to provide a full picture.

A good SSM should generalize well from training examples to unseen examples and be able to describe any valid instance of the shape class. Given an unseen test cohort of correspondence point sets, denoted $\mathcal{D}_{test}$, the **generalization** metric is defined as:

$$\text{Gen.} = \frac{1}{|\mathcal{D}_{test}|} \sum_{\mathbb{C} \in \mathcal{D}_{test}} ||\mathbb{C} - \hat{\mathbb{C}}||_2^2 \qquad (4)$$

where $\hat{\mathbb{C}}$ is the point set reconstructed via the training cohort PCA eigenvalues and vectors that preserve 95% variability. A smaller average reconstruction error indicates that the SSM generalizes well to the unseen test set.

Finally, effective SSM is specific, generating only valid instances of the shape class presented in the training set. This metric is quantified by generating a set of new sample correspondence points, denoted $\mathcal{D}_{sample}$, from the SSM generated on the training cohort, denoted $\mathcal{D}_{train}$. The **specificity** metric is quantified as:

$$\text{Spec.} = \frac{1}{|\mathbf{D}_{\text{sample}}|} \sum_{\mathbb{C}' \in \mathbf{D}_{\text{sample}}} \min_{\mathbb{C} \in \mathcal{D}_{train}} ||\mathbb{C}' - \mathbb{C}||_2^2 \tag{5}$$

The average distance between correspondence points sampled from the training SSM and the closest existing training correspondence points provides the specificity metric. A small distance suggests the samples match the training distribution well, indicating the SSM is specific.

## D  MODEL HYPERPARAMETERS

Model hyperparameters are provided in the following tables and in the configuration files proved with the code. Table 2 displays the hyperparameters that are consistent across all models. Note in the sparsity robustness experiments, the value of $N$ varies. Tables 3 and 4 display the hyperparameters specific to the CPAE and DPC models. These values match those originally tuned/reported. Finally, table 5 displays the hyperparameters specific to our Point2SSM model. The number of neighbors used in the ME loss is set to 10, as it is for DPC . The value of $\alpha$ is determined via tuning based on the validation set performance.

Table 2: Hyperparameters shared by all models.

| Shared Hyperparameters | | |
|---|---|---|
| Parameter | Description | Value |
| $N$ | Number of input points | 1024 |
| $L$ | Number of per-point features output by encoder | 128 |
| $M$ | Number of output points | 1024 |
| $B$ | Batch size | 8 |
| LR | Learning rate | 0.0001 |
| ES | Early stopping patience (epochs) | 100 |
| $\beta_1$ | Adam optimization first coefficient | 0.9 |
| $\beta_2$ | Adam optimization second coefficients | 0.999 |

Table 3: Hyperparameters specific to the CPAE model.

| Additional CPAE Hyper Parameters | | |
|---|---|---|
| Parameter | Description | Value |
| $\lambda_{MSE}$ | MSE loss weight | 1000 |
| $\lambda_{CD}$ | CD loss weight | 10 |
| $\lambda_{EMD}$ | EMD loss weight | 1 |
| $\lambda_{cc}$ | Cross-construction loss weight | 10 |
| $\lambda_{unfold}$ | Unfolding loss weight | 10 |
| $e$ | Adaptive loss epoch | 100 |

Table 4: Hyperparameters specific to the DPC model.

| Additional DPC Hyperparameters | | |
|---|---|---|
| Parameter | Description | Value |
| $K$ | Neighborhood size for loss calculation | 10 |
| $\gamma$ | Mapping loss neighbor sensitivity | 8 |
| $\lambda_{cc}$ | Cross-construction loss weight | 1 |
| $\lambda_{sc}$ | Self-construction loss weight | 10 |
| $\lambda_m$ | Mapping loss weight | 1 |

Table 5: Hyperparameters specific to our Point2SSM model.

| Additional Point2SSM Hyperparameters | | |
|---|---|---|
| Parameter | Description | Value |
| $K$ | Neighborhood size for ME loss | 10 |
| $\alpha$ | ME loss weight | 0.1 |

# E    MODEL MEMORY COMPARISON

Table 6 shows a comparison of the memory footprint of each model.

Table 6: Model memory footprint comparison. Size is reported in MB.

| Model | Total params | Forward/backward pass size | Params size | Total size |
|---|---|---|---|---|
| PN-AE | 3,832,576 | 11.04 | 14.62 | 25.68 |
| DG-AE | 4,702,336 | 609.54 | 17.94 | 627.5 |
| CPAE | 156,652 | 19.58 | 0.60 | 56.18 |
| ISR | 1,962,208 | 4.00 | 7.49 | 11.51 |
| DPC | 962,176 | 609.50 | 3.67 | 613.21 |
| Point2SSM | 22,098,560 | 633.69 | 84.3 | 718.01 |

Point2SSM has many more parameters than the other models due to the attention module. When this module is replaced with a simple MLP, as is done in the ablation experiment in appendix A, the number of parameters is reduced from 22,098,560 to 2,707,328. It is worth noting that this reduced model still provides more accurate SSM than the comparison models, as can be seen by comparing the metrics reported in table 1 row 3 and figure 3. This suggests that the performance improvement provided by Point2SSM is not simply a result of increased model capacity.

# F    BATCH SIZE ABLATION EXPERIMENT

The Point2SSM loss (equation 3) contains ME ( equation 2), which is computed pairwise within a batch. To analyze the impact of batch size on model performance, we perform an ablation on the pancreas dataset with batch sizes: 2, 4, 6, 8, 10, and 12. The results are provided in table 7. The batch size does not have a large impact on accuracy.

Table 7: Effect of batch size on Point2SSM performance on the pancreas dataset. Average values across test set are reported with best values marked in bold.

| | Point Accuracy Metrics (mm) ↓ | | | SSM Metrics ↓ | | | |
|---|---|---|---|---|---|---|---|
| Batch Size | CD | EMD | P2F | Comp. | Gen. | Spec. | ME |
| 2 | 2.74 | 1.42 | 0.279 | 24 | 2.12 | 4.41 | 3.38 |
| 4 | 2.61 | 1.41 | 0.256 | 23 | 2.22 | 4.61 | 3.22 |
| 6 | 2.64 | 1.41 | 0.270 | 23 | 2.20 | 4.65 | 3.25 |
| 8 | 2.72 | 1.42 | 0.283 | 24 | 2.15 | 4.55 | 3.36 |
| 10 | 2.67 | 1.42 | 0.269 | 24 | 2.14 | 4.66 | 3.32 |
| 12 | 2.71 | 1.42 | 0.280 | 24 | 2.14 | 4.59 | 3.34 |

# G    SHAPE DATASET VISUALIZATION

Figure 8 displays example shapes from each organ dataset (spleen, pancreas, and left atrium) from multiple views, illustrating the large amount of variation in these shape cohorts.

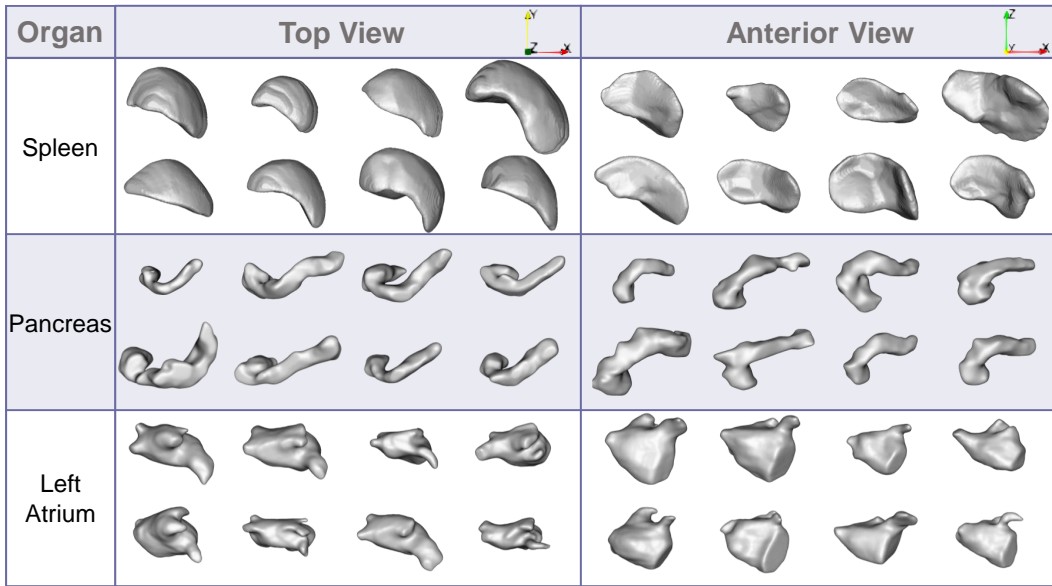

Figure 8: Example shapes are shown from each of the datasets from the top and anterior views.

## H  DOWNSTREAM EVALUATION: PANCREAS TUMOR SIZE CLASSIFICATION

As SSM is useful in many downstream applications, we have performed an additional evaluation of such a task. The pancreas dataset (Simpson et al., 2019) is comprised of cancerous cases, where each pancreas shape has tumorous masses of various sizes. As a downstream classification task, we analyze whether the tumor region of a given pancreas is larger or smaller than 20% of the pancreas size. In particular, we compute PCA embeddings of the SSM generated from each method. These PCA embeddings then serve as input into a random forest ensemble comprised of decision tree classifiers. The classifier is fit and evaluated using the original pancreas train/test split. The accuracy of the classifier built from each SSM is reported in table 8. The PSM and the comparison methods all performed similarly in this task, but Point2SSM provided a slight improvement. This evaluation demonstrates that Point2SSM provides effective, usable SSM estimation.

Table 8: **Pancreas Tumor Mass Classification Accuracy** Percentage of test cases correctly classified as having tumor regions are larger than 20% of the total pancreas size. Random forest ensemble classifiers were trained in the PCA embedding of the correspondence points provided by each method.

| Model | PSM | PN-AE | DG-AE | CPAE | ISR | DPC | Point2SSM |
|---|---|---|---|---|---|---|---|
| Accuracy | 85.71% | 85.71% | 85.71% | 82.29% | 85.71% | 85.71% | 89.28% |

## I  SPLEEN AND LEFT ATRIUM MODES OF VARIATION

Figure 9 displays the first two modes of variation captured by Point2SSM and all comparison models on the spleen and left atrium datasets. The CPAE results are excluded from this visualization because the resulting SSM is uninterpretable. The meshes in figure 9 are constructed using the output correspondence points. Mesh artifacts or implausible morphologies are an indication of output miscorrespondence. Point2SSM is the only point-based method that provides similar, if not more smooth and interpretable, modes to the PSM method.

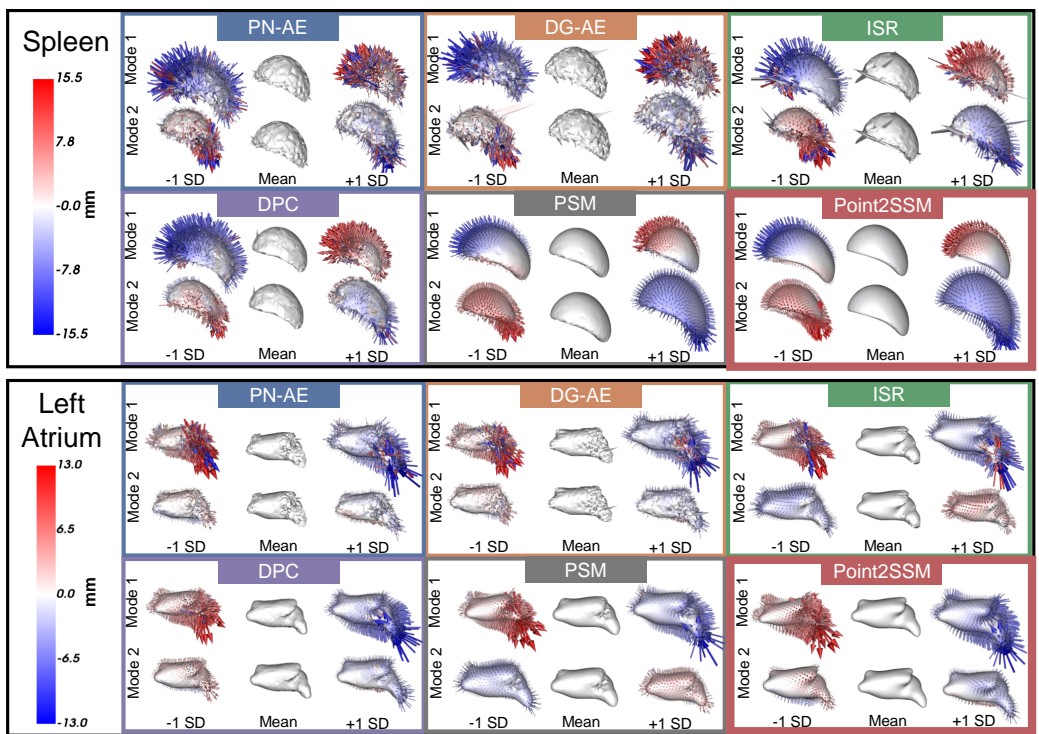

Figure 9: Primary and secondary modes of variation captured by spleen and left atrium SSMs output by each model. Point2SSM provides the smoothest and most plausible mean and modes of variation.

## J    ROBUST EVALUATION INPUT EXAMPLES

Figure 10 shows examples of input point clouds from the robustness experiments described in section 4.2.

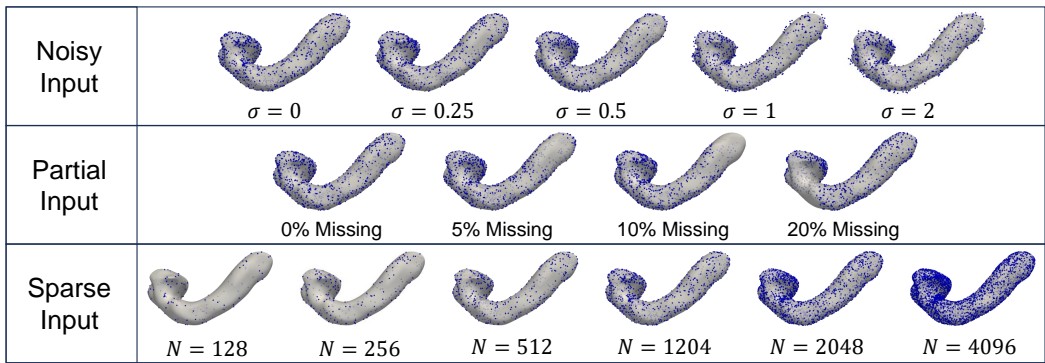

Figure 10: Robustness evaluation input point cloud examples are displayed over the semi-transparent ground truth shape.

## K    VERTEBRAE EXPERIMENT

In addition to the three organ experiments presented in section 4, we provide an experiment on a complex bone: the fourth lumbar (L4) vertebrae. This dataset is selected from the publicly available labeled and segmented data for human vertebrae by the vertebrae segmentation challenge (VerSe)

Sekuboyina et al. (2021). The L4 vertebrae data comprises 160 complete bone shapes. A visualization of a subset of the bones is provided in figure 11.

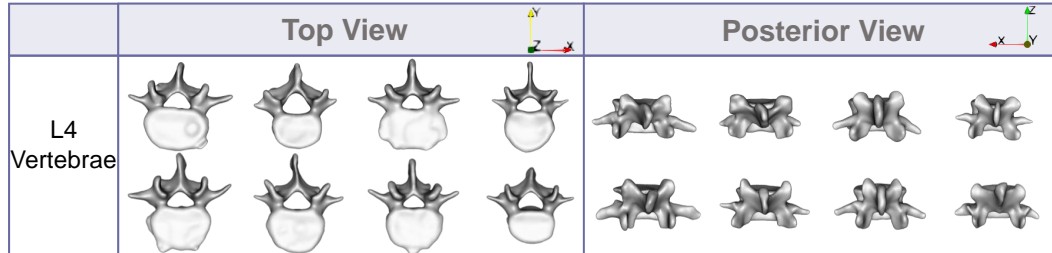

Figure 11: Example shapes are shown from each of the datasets from the top and anterior views.

The experiment was conducted using the same steps and parameters as described in section 4. The results are summarized in table 9. While PN-AE and DG-AE provide the most compact SSM's, they do not perform well on point accuracy metrics. CPAE performs poorly on all metrics, likely because the vertebrae topology is difficult to map to a sphere. ISR and DPC perform slightly better than the autoencoder models in terms of point accuracy, but suffer on the SSM metrics. Point2SSM performs best overall and provides an interpretable shape model with good correspondence and smooth modes of variation, as shown in figure 12.

Table 9: Average results on the L4 vertebrae test set. Best values are marked in bold.

| Model | Point Accuracy Metrics (mm) ↓ | | | SSM Metrics ↓ | | | |
| | CD | EMD | P2F | Comp. | Gen. | Spec. | ME |
|---|---|---|---|---|---|---|---|
| PN-AE | 5.22 | 1.82 | 0.897 | 24 | **0.606** | 1.70 | 2.03 |
| DG-AE | 4.90 | 1.79 | 0.836 | **22** | 0.615 | **1.67** | 2.09 |
| CPAE | 8.13 | 1.84 | 0.946 | 117 | 27.4 | 24.7 | 531.42 |
| ISR | 4.66 | 1.69 | 0.714 | 54 | 1.24 | 2.59 | 3.02 |
| DPC | 4.82 | 1.63 | 0.573 | 94 | 1.83 | 3.04 | 4.71 |
| Point2SSM | **2.61** | **1.48** | **0.304** | 34 | 0.879 | 2.06 | **1.98** |

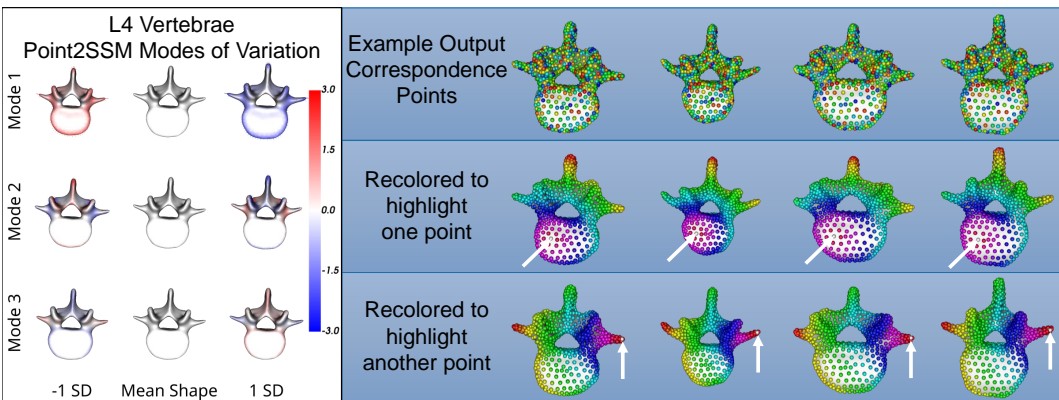

Figure 12: The L4 vertebrae SSM from Point2SSM is displayed. Left: The first three modes of variation are shown for the Point2SSM model at ±1 standard deviation from the mean. The heatmap and vector arrows display the distance to the mean. Right: Example predictions are shown where the point color denotes correspondence. Recoloring according to the distance to a selected point is provided for further illustration.

## L    MULTI-ANATOMY TRAINING

Table 10 compares the results of training and testing Point2SSM on a single anatomy versus multi-anatomy training. The results are similar, indicating multi-anatomy training neither helps nor hinders Point2SSM. Future work could leverage anatomy labels to increase the training signal and provide improved performance given multiple anatomies.

Table 10: Multi-anatomy Experiment

| Anatomy | | Point Accuracy Metrics (mm) ↓ | | | SSM Metrics ↓ | | | |
|---|---|---|---|---|---|---|---|---|
| Train | Test | CD | EMD | P2F | Comp. | Gen. | Spec. | ME |
| Spleen | Spleen | 3.42 | 1.53 | 0.363 | 8 | 4.19 | 6.11 | 3.25 |
| Spleen, Pancreas, Left Atrium | Spleen | 3.33 | 1.54 | 0.310 | 8 | 4.42 | 6.13 | 2.94 |
| Pancreas | Pancreas | 2.72 | 1.42 | 0.283 | 24 | 2.15 | 4.55 | 3.36 |
| Spleen, Pancreas, Left Atrium | Pancreas | 3.10 | 1.44 | 0.352 | 24 | 2.11 | 4.21 | 3.45 |
| Left Atrium | Left Atrium | 2.03 | 1.25 | 0.221 | 20 | 1.90 | 4.13 | 3.45 |
| Spleen, Pancreas, Left Atrium | Left Atrium | 2.15 | 1.25 | 0.258 | 20 | 1.86 | 4.09 | 3.47 |

