# OpenReview forum: "Point2SSM: Learning Morphological Variations of Anatomies from Point Clouds"
_ICLR.cc/2024/Conference — ICLR 2024 spotlight_

### Official Review · Reviewer_ARFH · 2023-10-28

**Soundness:** 4 excellent
**Presentation:** 3 good
**Contribution:** 3 good
**Rating:** 8
**Confidence:** 4

**Summary:**

The paper proposes a new model for learning statistical shape models (SSM) from unregistered 3D point cloud data for modelling anatomical shapes. The main contributions lie within the newly proposed architecture that enables attention based correspondences and later on more compact models, even from noisy or partial inputs.

**Strengths:**

The paper addresses an interesting and relevant problem that has not received enough attention in research yet. Shape models are very important for interpretable segmentation outputs.
The motivation and differentiation to related work is clear and the numerical and visual evaluation convincing. The visualisations are well done.
The paper discusses different mapping algorithms in detail and experimentally validates the positive aspects of the proposed global correspondence finding.

**Weaknesses:**

While the used metrics cover a good range of aspects from (Chamfer) surface distance to compactness and noise robustness, as well as a shape classification task in the supplement, other point distances e.g. density or rasterisation-based could have been explored.
The (compared) models have very many different aspects making a pin-pointing of the decisive difference harder. E.g. why did the authors not compare always the same geometric convolutional backbone / encoder for both correspondence and auto encoder-based solutions?
The method requires the computation losses of all pair-wise alignments within one mini-batch, which is computationally expensive but also raises the question why the authors did not directly compare to groupwise point registration as another good baseline.

**Questions:**

I wonder whether DPC is the optimal choice for a pairwise/groupwise correspondence model, as it does not consider instance optimisation which was shown to improve point registration nor does it create an unbiased mean for the shape model (but simply selects one reference cloud).

---

> ### Author Response · Authors · 2023-11-14
> **Response to Reviewer ARFH and request for clarification**
>
> Thank you for your thoughtful and constructive review of our paper.  We appreciate your feedback and the time and effort you dedicated to evaluating our work. We are pleased that you found the problem we addressed interesting and relevant. Your acknowledgment of the clarity in motivation and differentiation from related work is encouraging, and we are glad that the numerical and visual evaluations were convincing.
>
> We’d like to respond to each of your points and ask for some additional clarification.
>
> Regarding your suggestion to include additional point metrics such as density or rasterization-based, which metrics specifically would aid in evaluation? For Point2SSM and comparison models, we define m=1024 output correspondence points. The chamfer-based loss is two-sided, encouraging full coverage of the shape with well-spread points. As such, the correspondence point density is similar across models. Methods for rasterizing point clouds are a function of the specified resolution and suffer given shape with thin structures and highly convoluted shape details. The surface sampling metrics (including Chamfer and point-to-surface distance), along with the SSM metrics, provide a comprehensive evaluation of performance. Moreover, these are the same metrics used in similar work [1].
>
> We recognize Point2SSM and the comparison models have many different aspects and it is difficult to differentiate. In the revision, we will make the differences more clear. When comparing against existing methods, we implemented the models exactly as they were proposed for fair comparison. For this reason, we did not alter the backbones of the models. We did include a version of the autoencoder model with the dynamic graph convolution (denoted DG-AE) to aid in understanding the impact of the encoder architecture. Furthermore, we provide an ablation experiment in the supplementary material (Appendix A) to demonstrate the impact of each aspect of Point2SSM. This comparison includes encoder architecture (PointNet vs DGCNN), attention module architecture (self-attention vs MLP), and mapping error loss regularization (weighted by alpha). This illustrates how every design choice works together to provide the best accuracy.
>
> We’d like to clarify that the pairwise mapping error loss regularization term is not a registration loss. Rather, the mapping error encourages the point neighborhoods in one point cloud to be similar to those in another. This encourages correspondence directly, as point neighborhoods are well preserved in good correspondence models. We will clarify this in the revision.  Registration can be used as a surrogate task to correspondence, as accurate registration requires accurate correspondence points; however, we do not explore this in this work. Could you point us to the groupwise registration correspondence method you are referring to? This would enable us to address your concern better.
>
> Finally, we’d like to address your question about the optimality of DPC for groupwise correspondence. As there is no notion of ground truth correspondence points in these datasets, the creation of an unbiased mean isn’t possible. Instead, we select the point cloud with minimum Chamfer distance to all other point clouds in the training dataset to be the reference. This can be considered the medoid of the dataset. We found experimentally that the choice of the reference does not greatly impact accuracy, but the medoid is the least biased choice. We are unfamiliar with the notion of instance optimization in this context and would appreciate it if you could point us to some references.
>
> Thank you again for the detailed and constructive feedback. If you could provide further detail regarding additional point distance metrics, groupwise registration correspondence methods, and instance optimization for DPC, we would appreciate it. That would enable us to respond more thoroughly to your feedback and improve the paper appropriately.
>
> [1] Adams, Jadie, and Shireen Y. Elhabian. "Can point cloud networks learn statistical shape models of anatomies?." International Conference on Medical Image Computing and Computer-Assisted Intervention. Cham: Springer Nature Switzerland, 2023.

---

### Official Review · Reviewer_gxDE · 2023-11-04

**Soundness:** 3 good
**Presentation:** 3 good
**Contribution:** 2 fair
**Rating:** 5
**Confidence:** 4

**Summary:**

This paper addresses a clinically relevant problem of statistical shape modeling from point cloud data of different anatomical structures. Specifically, the authors learn to reconstruct the point cloud and learn correspondence in a self-supervised manner. The results show improved SSM on three anatomical structures.

**Strengths:**

Strength:

+ The paper is well-motivated and has done a thorough literature review.

+ The results improve SSM performance over previous methods.

+ The method shows robustness against noisy, partial, and sparse input.

**Weaknesses:**

Major comments:
- Although the paper addresses a clinically relevant application, the technical novelty of this paper is limited since it is based on existing architecture and loss without any new insights or analysis. This remains a major weakness and is hard to address during rebuttal.
- Since the corresponding promoting loss acts between the minibatch samples, what is the effect of minibatch size on the model performance?
- Only three anatomical structures have been studied. The inclusion of more anatomical structures is needed to support the generalizability of the method.
- The methods describe the objective as reconstructing a sparse set of point reconstruction but, in practice, uses N=M=1024, which raises concern about the initial objective.

**Questions:**

see weaknesses

---

> ### Author Response · Authors · 2023-11-16
> **Response to Reviewer gxDE**
>
> We appreciate your thoughtful review and are grateful for the constructive feedback. We have carefully considered your comments and suggestions to improve the quality of our work.
>
> We acknowledge your concern regarding the technical novelty of our paper, particularly in terms of not introducing a novel loss function or network layers. While it's true that our work draws inspiration from various tasks and fields (such as keypoint detection, self-attention, and pairwise point correspondence), we want to emphasize the uniqueness of our contribution lies in the synthesis of these ideas for the specific task of anatomical SSM. Point2SSM is the first deep learning method designed to directly generate 3D anatomical statistical shape models from point clouds in an unsupervised manner.  Although individual components have precedent in previous work, the novel combination and resulting overall architecture, as illustrated in Figure 2, is a distinctive contribution. Our results demonstrate the advantages this innovative approach offers. Our work provides significant value as it enhances the feasibility of SSM generation and broadens its possible applications, potentially accelerating its adoption as a clinical research tool.
>
> Regarding your inquiry about the impact of minibatch size on model performance, we have conducted an ablation experiment on the pancreas data in the revised paper, see Appendix J. This ablation demonstrates that Point2SSM accuracy is not sensitive to batch size.
>
> While we initially focused on three highly variable anatomical structures due to space constraints, we understand your suggestion for including more structures to convey generalizability. To address this, we have introduced an additional experiment on a vertebrae dataset in Appendix K, featuring a complex bone with thin structures that serves as a challenging test case. This supplementary experiment further supports the improved accuracy and generalizability demonstrated by Point2SSM.
>
> Finally, regarding your inquiry about sparse reconstruction, we’d like to clarify the objective of Point2SSM is to predict $M$ correspondence points from $N$ unordered, randomly selected input points. The $M$ correspondence points should be well-spread and dense enough to capture the details of the shape. While related work in computer vision has sought to learn sparse keypoints or structure points, sparsity is not a goal in SSM. In our main experiments, we elect to use $N=M=1024$ as this ensures the output points are dense enough to capture the complexity of the organs. We also performed an input sparsity ablation experiment with varying input sizes (from $N=128$ to $N=4096$), and the results are depicted in the "Sparse Input" row of Figure 6. This analysis affirms that Point2SSM outperforms comparison models across different input sizes. In our revision, we will provide additional justification for this choice and clarify any potentially misleading statements about sparse reconstruction.
>
> We appreciate your attention to detail and are committed to addressing all your comments to enhance the overall quality of our paper. We hope this explanation and the addition of Appendices J and K help address your concerns. Please let us know if you have remaining questions.

---

### Official Review · Reviewer_PV7h · 2023-11-06

**Soundness:** 3 good
**Presentation:** 3 good
**Contribution:** 3 good
**Rating:** 8
**Confidence:** 3

**Summary:**

The author introduces Point2SSM, a novel unsupervised learning approach designed for constructing statistical shape models (SSMs) from point clouds, a method crucial for analyzing morphological variation in anatomy in clinical research. Traditional SSM creation methods have limitations, such as the need for noise-free meshes, reliance on assumptions, and long inference times. Point2SSM addresses these issues by inferring SSMs from raw point clouds, a process that is more efficient and widely applicable due to the ease of acquiring point cloud data. Despite recent advances in deep learning for 3D point clouds, the construction of SSMs for anatomy faces challenges such as noisy or incomplete input and limited training data. Point2SSM overcomes these by using an attention-based module for correspondence mappings, demonstrating superior performance in surface sampling and correspondence, thus enhancing the capture of population-level anatomical statistics.

**Strengths:**

Point2SSM introduces a novel unsupervised learning framework that directly constructs SSMs from point clouds, a significant advancement over traditional methods. Additionally, this method addresses the limitations of classical SSM generation methods, such as the need for noise-free surface meshes and reliance on predefined templates. By incorporating local information, it utilizes a DGCNN encoder to learn features by incorporating local neighborhood information, capturing global semantic characteristics of anatomical shapes. Moreover, the attention module in Point2SSM predicts correspondence maps in a self-supervised manner, eliminating the need for labeled data. The results show robustness against noisy, partial, and sparse inputs, which is critical for clinical modeling.

**Weaknesses:**

Single Anatomy Modeling: Point2SSM's current limitation to model only one anatomical structure at a time could be expanded upon. How might the method be adapted to accommodate the study of multiple anatomical structures simultaneously, and could this expansion improve the model's understanding of the interrelationships between various anatomical features?

Pre-alignment Requirement: The necessity for pre-aligned input point clouds could restrict Point2SSM's use in scenarios with non-aligned data sources. What methods could be integrated into Point2SSM to automate the alignment process, or to make the model robust to variations in data orientation due to patient movement or diverse scanning protocols?

Uncertainty Quantification: The lack of uncertainty quantification in the model's outputs is a notable concern for clinical decision-making. Are there potential strategies for integrating uncertainty estimation into Point2SSM to make it more suitable for clinical environments where risk assessment and error margins are vital?

Performance with Sparse Data: While Point2SSM has demonstrated promising results with limited datasets, its performance with sparse data is not fully explored. Investigating this could provide insights into how the model might be optimized to better capture the variability and complexity of anatomical shapes, leading to more accurate and robust SSMs.

Model Expansion for Multiple Anatomies: Enhancing Point2SSM to handle misaligned inputs and model multiple anatomies could greatly increase its clinical relevance. How could the model be developed to not only process various anatomical structures within a single framework but also address the variations and interconnections between them?

**Questions:**

How could Point2SSM be modified to model multiple anatomical structures concurrently and understand their interrelationships?
What approaches could be integrated into Point2SSM to allow for automated pre-alignment or to make the model resilient to unaligned data from diverse orientations?
Are there methods to incorporate uncertainty quantification into Point2SSM's outputs to aid risk assessment in clinical decision-making?
How could Point2SSM be optimized for improved performance with sparse data, ensuring more accurate and robust statistical shape models?
In what ways could Point2SSM be developed to handle both misaligned data and the modeling of multiple anatomies within a single framework?

---

> ### Author Response · Authors · 2023-11-16
> **Response to Reviewer PV7h**
>
> We appreciate your review of our work and thank you for your constructive feedback. The weaknesses you have highlighted are important considerations, all of which we have duly addressed in Section 4.3 "Limitations and Future Directions."  Point2SSM is the first unsupervised deep learning method designed to generate 3D anatomical statistical shape models from point clouds. This work lays the foundation for future endeavors aimed at addressing the identified limitations.
>
> We’d like to address each of your questions:
> - Single Anatomy Modeling: Point2SSM is designed to replace traditional SSM construction methods, such as PSM, which model only a single anatomy. For this reason, experiments were performed on a single anatomy, allowing Point2SSM to learn anatomy-specific features. However, Point2SSM is not inherently restricted to a single anatomy.  Our revised paper (see Appendix L) presents an experiment showcasing Point2SSM's performance when trained on multiple anatomies, yielding comparable results to a single anatomy model. In future work, Point2SSM could be modified to utilize anatomy labels in multi-anatomy training to potentially leverage both anatomy-specific and common properties. This would require a more sophisticated approach and is out of the scope of this work.
> - Pre-alignment Requirement: Existing approaches for building SSMs require shapes to be pre-aligned, and there are available tools for doing so, including center of mass and iterative closest point alignment. Using aligned input in Point2SSM allowed for direct comparison with the optimization-based PSM approach. This was important in the evaluation as it allowed us to compare the correspondence accuracy of Point2SSM and the other point cloud deep learning techniques to a traditional optimization-based method. Future work could explore extending Point2SSM to allow for misaligned input. However, this would require performing post-alignment on the predicted correspondence points so that anatomy orientation and location are not captured as modes of variation in the resulting SSM.
> - Uncertainty Quantification: Point2SSM, like all established methods of its kind, is a deterministic neural network that does not quantify uncertainty. You are correct that this is an important future direction for this work, as it would improve the trustworthiness of output in a clinical scenario. Many existing methods of uncertainty quantification in neural networks could be directly applied to Point2SSM, such as Monte Carlo variational dropout [1], concrete dropout [2], ensembling [3], etc. Progress has been made in uncertainty quantification in supervised deep learning methods for predicting SSM from images [4]. These methods could be extended to Point2SSM; however, it is out of the scope of this work.
> - Performance with Sparse Data: We have performed an experiment evaluating performance given sparse data, the results of which are in the “Sparse Input” row of Figure 6. Here, we tested the ability of Point2SSM and comparison models to predict 1024 output correspondence points given sets of input points of size 128, 256, 512, 1024, 2048, and 4096. Point2SSM performs the best overall, given all of these input sizes. To aid in understanding this comprehensive evaluation, we’ve added a visualization that showcases examples of sparse input points at each level in Appendix I.
> - Model Expansion for Multiple Anatomies: We agree that enhancing Point2SSM to handle misaligned inputs and model multiple anatomies would increase its usability. We have described this as a future direction to explore. Even without these features, Point2SSM greatly enhances the feasibility of SSM generation and broadens its possible applications. The abundant future directions of Point2SSM indicate that this contribution will be of value and interest to the community.
>
> In addition to the inclusion of Appendices I and L, we will expand the "Limitations and Future Directions" section of the paper in the revision. While Point2SSM has inherent limitations, it presents substantial advantages over traditional SSM generation approaches and surpasses state-of-the-art point cloud networks. Please let us know if you have any remaining questions or concerns.
>
>
>
> [1] Gal, Yarin, and Zoubin Ghahramani. "Dropout as a Bayesian approximation: Representing model uncertainty in deep learning." *International conference on machine learning.* PMLR, 2016.
>
> [2] Gal, Yarin, Jiri Hron, and Alex Kendall. "Concrete dropout." *Advances in neural information processing systems* 30 (2017).
>
> [3] Lakshminarayanan, Balaji, Alexander Pritzel, and Charles Blundell. "Simple and scalable predictive uncertainty estimation using deep ensembles." *Advances in neural information processing systems* 30 (2017).
>
> [4] Adams, Jadie, and Shireen Y. Elhabian. "Fully Bayesian VIB-DeepSSM." *International Conference on Medical Image Computing and Computer-Assisted Intervention.* Cham: Springer Nature Switzerland, 2023.

---

### Official Review · Reviewer_x8Fq · 2023-11-09

**Soundness:** 3 good
**Presentation:** 3 good
**Contribution:** 3 good
**Rating:** 8
**Confidence:** 4

**Summary:**

This paper proposes a method, Point2SSM, for the task of correspondence-based statistical shape modeling(SSM), This refers to mapping a shape to a set of $M$ keypoints (correspondence points (CPs) in the paper). The CPs must accuractely summarize a shape and correspond to consistent anatomical features such that they can be used, e.g., with PCA, to compute population level statistics.

Point2SSM maps $N$ input points to $M$ CPs. It consists of a dynamic graph CNN (DGCNN) encoder which outputs a feature vector for each input point. Then, self-feature augment (SFA) attention is applied on the feature vectors. Softmax is then applied to produce an $M x N$ attention matrix that maps the input points to the CPs. Point2SSM is trained in an unsupervised fashion. A Chamfer loss is used to ensure the CPs accurately summarizes the input shape and an adapted pairwise mapping error (ME) is used to ensure the CPs correspond to consistent features. While each components of Point2SSM does not seem novel, their combination is.

The authors compare Point2SSM with several recent and relevant methods on three datasets of spleens, pancreases, and left atriums. They show that Point2SSM summarizes shapes significantly better than the compared methods, while being competitive regarding the consistency of the CPs. They perform additonal experiments showing improved robustsness and include ablation experiments in the supplementary that shows the benefit of each component.

**Strengths:**

- The paper is well written. It is well motivated and it is clear how the method works and how the experiments are set up.
- The results show clear improvements over the compared approaches.
- The experiments are comprehensive enough to demonstrate the benefits of Point2SSM.
- The use of the attention module provides clear benefits - this is a nice finding.

**Weaknesses:**

- The method seems to have significant overlap with the ISR method from Chen et al., 2020. Specifically, it seems to me that replacing PointNet++ with DGCNN + SFA in ISR would almost be the same method. This is not an issue for me, but: if true, I think the inspiration/similarities should be mentioned and Chen et al., 2020 cited in sec. 3.1. If false, I would like the authors to clarify the differences.
- It is stated that enforcing the CPs to be a convex combination of the input points "increases surface sampling accuracy". It is not fully clear to me why this must be true. From the attention maps in App. B, Fig. 7, right, CPs could be placed very inacurately even if they are convex combinations of the attended points. I would like clarification on this point. If it is an empircal observation, I would like it clearly stated (ideally, with an ablation experiment but this is not crucial to me).
- It is not clear to me why Point2SSM has such a benefit when it comes to adding new samples. How would this work? Would one "simply" continue training from previous weights, or does the architecture enable something smarter?
- Minor point: At the end of sec. 2.3, it is stated that: "These methods establish correspondence but are prone to overfitting given a limited training budget and are not robust to noise, missingness, and sparsity in the input point cloud." I find this to be a strong statement. Could it be softened or further justified?

**Questions:**

- Point2SSM has many more parameters than other models (App. E, Table 6.). Is comparing it with the much smaller models fair? Would "upscaling" the other models narrow the performance gap? I would like the authors to clarify why/why not.
- Top of page 5: please specify what "Euclidean neighbors" mean. As I can tell, it is the K-nearest neighbors - if so I suggest using that. While Lang et al., 2021 also used "Euclidean neighborhood", I think  it would make the text in this paper easier to read with another term.
- The text in the figures is very small. Also, blue text on blue background in Fig. 5 is hard to read. Improving this would make the results much easier to interpret.
- For the robustness experiments, it would be nice to illustrate how the pertubations look like. This can just be in the supplementary.
- For the sparse input experiment, how are the points sampled? Random, farthest point samping or other?
- When illustrating modes of variation, I find it more clear to deform the mean shape instead of showing deformation vectors. However, this is minor for me and the authors may choose to change it or not.
- Section 3.1, first line: second use of "unordered" seems redundant.

---

> ### Author Response · Authors · 2023-11-16
> **Response to Reviewer x8Fq**
>
> Thank you for your thorough review of our paper. We are pleased that you found the paper well-written and the experiments comprehensive, and we are grateful for the constructive critiques that will undoubtedly improve our work.
>
> We would like to address your specific comments and questions:
> - Overlap with ISR Method: Our method is similar to, and partially inspired by, the ISR method from Chen et al., 2020. However, in addition to the use of DGCNN and SA instead of PointNet++, the overall architecture is different. In ISR, a PointNet++ encoder learns a subset of points, called sample points, and associated features. The MLP-based “Point Integration Module” then learns to transform these sample points to output correspondence points. Where Point2SSM learns a weighted combination of input points, ISR outputs a weighted combination of learned sample points. Additionally, Point2SSM employs the mapping error loss regularization term, whereas ISR utilizes Chamfer distance alone. We acknowledge these methods are similar and appreciate your suggestion to clarify this in Section 3.1. In revision, we will provide a detailed comparison, and acknowledge the similarities.
> - Convex Combination of Correspondence Points (CPs): The statement that requiring the CPs to be a convex combination of the input points increases surface sampling accuracy could be made more clear. While this factor does not constrain CPs to the surface directly, constraining them to be within the convex hull prevents them from being very far away. Empirically, we see that the methods with this constraint (Point2SSM, ISR, and DPC) perform better on the point accuracy metrics (Figure 3). We will make this statement more clear in the revision.
> - Benefit of Adding New Samples: In the paper we state: “Point2SSM can also enable sequential or online learning, as well as incremental model updating, making it highly adaptable to real-world clinical scenarios where shape data accumulates over time.” This is an advantage of the deep learning approach in general, as compared to the traditional optimization-based SSM construction methods. We will rephrase this to make sure it does not suggest that Point2SSM has a benefit over other deep learning methods in this regard.
> - Strong Statement about Established Methods: We agree that: "These methods establish correspondence but are prone to overfitting given a limited training budget and are not robust to noise, missingness, and sparsity in the input point cloud." is a strong statement. It is more appropriate to say that these methods have not been stress tested with regard to these conditions. We will revise the paper to soften the statement appropriately.
>
> Questions:
> - You raise an interesting question about the model scale. Point2SSM has many more parameters than the other models due to the attention module.  However, the ablation experiment in Appendix A helps demonstrate that the accuracy improvement is not simply a product of the model scale. The third row of Table 1 reports results when the attention module is replaced with a simple MLP, reducing the parameters from 22,098,560 to 2,707,328. The results, while worse than the full Point2SSM model, are still better than the comparison models performance on the pancreas dataset, as reported in Figure 3. This indicates that upscaling the other models would not close the performance gap. We will revise Appendix E to include this discussion.
> - “Euclidean neighbors” at the top of page 5 does mean K-nearest neighbors. Here, the term Euclidean neighbors was used to clarify that the nearest neighbors are found using Euclidean distance rather than geodesic distance, which would be preferred but requires a full mesh representation of shape. We will rephrase this to make it clear that it is K-nearest neighbors computed using Euclidean distance.
> - In revision, we will improve the legibility of the text in figures and correct the blue-on-blue text in Figure 5.
> - We agree that visualizations of the input perturbations would be helpful in understanding the robustness evaluation. We have already added this to the revised version, see Appendix I.
> - In the sparse input experiment, the points are sampled randomly. This task is more challenging than if points were sampled via farthest point sampling. We will clarify this in the revision.
> - When illustrating modes of variation, we are both deforming the mean shape and showing the deformation vectors to provide the clearest visualization of the differences.
> - We agree the second use of "unordered" is redundant in Section 3.1, and we have removed it.
>
> We sincerely appreciate your detailed and constructive feedback, and we are committed to addressing each point to improve the overall quality of the paper. If you have any further questions or concerns, please let us know.

---

> > ### Comment · Reviewer_x8Fq · 2023-11-19
> >
> > I would like to thank the authors for their detailed response to me and the other reviewers. I appreciate the authors will improve the clarity and legibility of the figures in a revision. I would be even further reassured if the authors could provide more details/suggestion regarding the text revisions planned. As far as I can see, they have not been included in the current revision.
> >
> > I am satisfied with the responses to my questions. I further think that the additions made in response to the other reviewers have further improved the paper. Unless the remaining reviewers raise further objections, my inclination is to keep my score.

---

> > > ### Author Response · Authors · 2023-11-20
> > >
> > > We appreciate your response and are pleased to have addressed your questions adequately. The current version has just been updated with additional clarifications in the text, highlighted in green. We've also enlarged the font size in Figures 1, 3, 4, and 5. All figures are vector-based, allowing readers to zoom in as necessary. We remain committed to refining the text within the page limit and will continue to improve it based on reviewer feedback.

---

### Author Response · Authors · 2023-11-16
**Summary Response**

We thank the reviewers for their time and valuable feedback, which will strengthen our work. We have responded to each individual question and concern. Based on the reviewer's feedback, we have added four appendix sections to the supplementary materials:
- **Appendix I** contains a visualization of perturbed input point cloud examples to aid in understanding the robustness evaluation in Section 4.2, as Reviewer x8Fq recommended.
- **Appendix J** contains an ablation experiment to demonstrate the effect of batch size on Point2SSM accuracy, as Reviewer gxDE recommended. This ablation demonstrates that Point2SSM is not sensitive to batch size.
- **Appendix K** contains an additional experiment on a new dataset. Reviewer gxDE noted that the inclusion of more anatomical structures would better support the generalizability of Point2SSM. We have added an experiment on the fourth lumbar vertebrae, a highly complex bone. The results are similar to the three organ experiments in the main paper. Point2SSM performed best overall, further supporting its superiority.
- **Appendix L** contains a comparison of results from training Point2SSM on single anatomy versus multiple anatomical structures. Point2SSM provides similar accuracy on all metrics given single or multi-anatomy training. This experiment aids in addressing Reviewer PV7h’s concern that Point2SSM is limited to single anatomy training.

These revisions to the paper have been added in green. Further revisions will be made to the text to clarify the concerns raised by reviewers as specified in the individual comments. Once again, we express our gratitude for the meticulous evaluation of our work. We believe Point2SSM is a significant contribution that paves the way for further exploration in SSM and correspondence in computer vision at large and will be of great interest to the community.

---

### Author Response · Authors · 2023-11-21
**General Comment and Friendly Reminder**

We have added additional clarification to the revised version of the paper, with changes marked in green. For reviewers who have not yet responded to our comments, we kindly remind you that the discussion period ends tomorrow. We would very much value your input on our revised manuscript and hope for the opportunity to address any remaining concerns.

---

### Author Response · Authors · 2023-11-22
**Gentle Reminder and Request for Review of Responses as Discussion Period Nears Conclusion**

As the discussion period swiftly approaches its conclusion, we kindly remind the reviewers to review our recent responses to your valuable comments and queries regarding our submission. We greatly appreciate the insights and constructive feedback you have provided thus far. In an effort to ensure that all your concerns are thoroughly addressed, we would be grateful if you could take a moment to examine our clarifications and let us know if any further information or detail is required. Your guidance is crucial in refining our work, and we are eager to make any necessary amendments to meet the high standards of the review process. We look forward to your feedback and are ready to provide any additional clarifications that may be needed.

---

### Meta-Review · Area_Chair_LSiM · 2023-12-05

**Metareview:**

This paper introduces Point2SSM, an unsupervised learning approach that can learn shape models from unregistered point clouds. The authors introduce an attention module that enables self-supervised learning of corresponding, and they show that their model is robust to noisy or incomplete data. Weaknesses include lack of uncertainty quantification as well as lack of ability to model multiple organs simultaneously. The authors acknowledge these weaknesses as future work, but also note that the existing state-of-the-art suffers from the same weaknesses.

**Justification For Why Not Higher Score:**

From the ICLR point of view I'm not sure the paper has sufficiently general interest to warrant an oral.

**Justification For Why Not Lower Score:**

As the reviewers point out, this is a paper that is carried out well and shows very convincing results in shape modelling.

---

### Decision · Program_Chairs · 2024-01-16

Accept (spotlight)